# Study on the Rehydration Quality Improvement of *shiitake* Mushroom by Combined Drying Methods

**DOI:** 10.3390/foods10040769

**Published:** 2021-04-03

**Authors:** Lina Hu, Jinfeng Bi, Xin Jin, Yang Qiu, R. G. M. van der Sman

**Affiliations:** 1Institute of Food Science and Technology, Chinese Academy of Agricultural Sciences (CAAS), Key Laboratory of Agro-Products Processing, Ministry of Agriculture and Rural Affairs, Beijing 100193, China; lina.hu@wur.nl (L.H.); qymbmh1995@163.com (Y.Q.); 2Food Biobased Research, Wageningen University & Research, Bornse Weilanden 9, 6708 WG Wageningen, The Netherlands; ruud.vandersman@wur.nl; 3Food Process Engineering, Agrotechnology and Food Sciences Group, Wageningen University & Research, Bornse Weilanden 9, 6708 WG Wageningen, The Netherlands

**Keywords:** serial combined drying, rehydration, mushroom, microstructure

## Abstract

The aim of study is to improve the rehydration quality of dried *shiitake* mushrooms for their instant food manufacturers and fast restaurants. Serial combined drying methods were investigated to achieve this objective: either instant controlled pressure drop drying (DIC) or freeze drying (FD) was used as the treatments for microstructure improvement, and they were applied either before or after an additional drying step at low (35 °C) or high (65 °C) temperatures. Dried mushrooms were assessed for quality indicators like relative volume, rehydration rate, dry matter loss and sensory scores. Microstructure properties were inferred to understand the physical mechanisms of quality changes. Principal component analysis (PCA) was used to cluster treatments and to identify combinations of drying techniques, rendering improved quality. Consequently, it was shown that DIC treatment before hot air drying at 35 °C was shown to be the most promising combined drying method to enhance the rehydration quality, leading to a high volume recovery ratio, low dry matter loss after rehydration, and high rehydration rates. This good performance could be explained by the retention of pore interconnectivity resulting from the slight expansion of porous structure during DIC and the retention of cell membrane integrity.

## 1. Introduction

Mushroom production in China has increased rapidly and accounts for over 70% of the world’s production (FAO, 2019). Among them, the *shiitake* mushroom is always the most commonly cultivated in China and is mainly processed by drying. Drying processes of *shiitake* mushrooms are widely investigated because drying not only prolongs their shelf life but also helps to generate their umami flavor [1,2]. In general, dried *shiitake* mushrooms are required to be rehydrated before being consumed. The ideal quality attributes of rehydrated products are expected to be high convenience (high rehydration rates) and similar appearance, texture and juiciness as fresh products. However, most studies have focused on the drying characteristics and quality attributes of dried products, while problems with rehydration properties have not been investigated thoroughly.

It is known that different drying methods and conditions can result in large quality differences of dried and rehydrated products [3,4,5,6,7,8,9]. As the major drying methods in industrial production, hot air drying (HA) and freeze drying (FD) have been investigated from both the academic and industrial points of view for many years. However, both methods still have problems in producing desired products with good rehydration quality. In particular, hot air drying (HA)leads to slow rehydration and tough texture [8,9]. While freeze-drying (FD) shows good rehydration and volume retention, texture is considered too fragile when eaten [10,11,12,13], which makes freeze-dried products less attractive for consumers. The rehydration performance of a dried food product is largely determined by the changes in food structure resulting from microscale deformation during drying [14,15,16]. An interconnected, porous structure is critical for good rehydration, and this can be obtained by freezing drying (FD), microwave drying, frying, and instant controlled pressure drop (DIC) [15,17,18].

From previous studies, combined drying methods shows advantages in improving drying times, rehydration and sensorial qualities of vegetable products [19,20,21,22,23], which could be a promising way of obtaining “ideal” rehydrated mushrooms. Generally, there are two kinds of combined drying methods: serial and parallel drying. Serial combined drying involves the use of one drying method followed by another. Parallel combined drying includes two or more drying methods implemented simultaneously. Notably, parallel combined drying requires specialized equipment, involving high investment and knowledge development for proper process design. Contrastingly, serial combined drying can easily be performed with existing drying equipment. 

Several earlier studies have investigated serial combined drying [23,24,25,26,27]. Siebert [23] found that FD as pretreatment before hot air or microwave-vacuum drying creates a porous surface layer, which prevents case hardening, and retains volume similar to FD. Furthermore, FD pretreatment accelerates overall drying time. However, HA pre-drying before FD promoted tissue collapse, which could not be undone by FD, rendering poor rehydration and prolonged drying time. Besides, because the least efficient part of a conventional drying system (hot air or freeze drying) is near the end of the process (the falling rate period), when most of the time may be spent removing the the small remaining portion of moisture content, an efficient drying method or treatment, such as infrared drying or instant controlled pressure drop (DIC), can be used at the second stage to reduce the drying time [20,28]. It can be concluded that the proper combination of drying methods needs to be designed carefully to improve both quality and drying efficiency.

This study therefore set out to investigate the proper combination of drying methods to obtain products with the desired rehydration quality. Based on the earlier experience of Siebert [22,23] and Mujumdar [21,29], we have devised the following serial combined drying strategy: one drying method (DIC or FD) is used to enhance the porous structure, while another conventional drying method is used to remove the water efficiently. The latter methods are applied either before or after the DIC or FD treatments, and they are chosen on the basis of their reported good rehydration performance and drying efficiency. Accordingly, in this study, we investigated the most promising combination of drying techniques that overcomes the disadvantages of single drying methods like HA or FD. Quality indicators linked to rehydration properties and sensorial attributes of shiitake mushrooms were measured. PCA analysis was applied to the large data set to obtain the most discriminating quality traits amongst treatments, and to cluster them for more in-depth analysis with regard to microstructural changes due to drying. From the PCA results and our subsequent analysis the most promising combined drying method was selected.

## 2. Materials and Methods

### 2.1. Experimental Design

Our experiments with serial combined drying are performed using either instant controlled pressure drop (DIC) or freeze-drying (FD) to enhance the porous structure. They are combined with the following conventional drying methods for moisture removal: (1) hot air drying (HA), (2) heat pump drying (HP), (3) vacuum drying (VD), and (4) infrared drying (IRD). These conventional drying methods were selected due to their practical or promising application in industry, where hot air drying methods are the most widely used for their low cost and ease of control; the heat pump and infrared drying methods have advantages in lower energy consumption or high drying efficiency, and they have been increasingly used as the technique for drying of agricultural products in recent years; vacuum drying is used when the product is heat-sensitive, and in our case we try to keep the cell intact so as to benefit the rehydration quality. These conventional drying methods were performed either before or after the DIC/FD treatments. Besides, the effect of the drying temperature of conventional drying techniques on the rehydration quality was assessed with two extreme temperatures, a higher temperature (65 °C) and a lower temperature (35 °C). The two temperatures were chosen based on the capacity of our equipment and assumed to give acceptable quality performance according to previous studies, whose drying temperature varies from 40 °C to 70 °C [30,31,32]. This made a total of 32 different treatments, and for convenient discussion we abbreviated the treatments as either predrying35/65-DIC/FD, or DIC/FD-postdrying35/65, with pre-drying and post-drying indicating that the (conventional) drying method is applied before or after the DIC/FD treatment, respectively. The results of the serial combined drying methods were compared to single drying techniques, namely hot air drying at 35 °C and 65 °C (denoted as HA35 and HA65, respectively), and freeze-drying (FD), which all serve as reference drying methods.

An important design parameter for the combined drying methods was the switch-over condition, when the mushroom was transferred between drying methods. For the porosity enhancing methods, DIC and FD, we used fixed treatment times, which were shorter than for the reference single drying treatments, which were 44 h for HA35, 16 h for HA65, and 48 h for FD. The chosen treatment times for the porosity enhancing methods were chosen as 3 h for DIC, and 8 h for FD in the case of the post-drying treatments. This resulted in 30% remaining moisture content (wet basis) for the FD treatment, and 80% remaining moisture content for the DIC treatment. For the pre-drying treatments, we had determined that the switch-over was to happen at 30% moisture content (wet basis), as earlier preliminary studies had shown a remarkable decrease of the drying rate at this point. The corresponding switch-over time was determined via preliminary drying experiments, where mushrooms were weighed each hour using a digital balance (Sartorius BSA223S, Germany). For the post-drying, we had determined the treatment times required to obtain a final moisture content of 13% (wet basis), which are also listed in Table 1.

Samples obtained from all drying treatments were evaluated for various quality traits after drying or after rehydration. The following quality indicators were measured: (1) the volume retention ratio after drying, (2) the volume recovery ratio after rehydration, (3) the rehydration rate, (4) the dry matter loss ratio during rehydration, (5) the water holding capacity (WHC), (6) the hardness and (7) the sensorial scores of cooked mushrooms. Other measurements targeted changes in microstructure: (8) the porosity after drying, (9) the porosity after rehydration, and (10) the cell membrane integrity. Because of the large amount of data generated we used principal component analysis (PCA) to determine the principal quality traits distinguishing all treatments, and to guide us in the discussion of the large number of results.

### 2.2. Materials

Fresh cultivated *shiitake* mushrooms were purchased from a mushroom farm in Beijing (China), where they were graded to have a uniform maturity and size. All mushroom samples were transported to the laboratory and used for experiments within 48 h after harvest. Before experiments, they were put temporarily in a cold store at a temperature of 4 °C and 95% relative humidity. The average dimension of the mushroom cap was 60.2 ± 3.1 mm. The initial moisture content of the whole mushroom was 85.2 ± 1.7% (wet basis: w.b.), as measured by heating triplicate samples in a drying oven at 105 °C for 24 h (AOAC. 1990). 

### 2.3. Drying Methods 

The applied drying methods were performed as follows:Freeze drying (FD): Fresh shiitake mushrooms were frozen in a refrigerator at −80 °C for 6 h, and then they were put in a lab-scale freeze dryer (Alpila-aplus 4Lplus, Christ Co. Ltd., Osterode am Harz, Germany). The corresponding drying stage of 24 h was performed at a chamber pressure of 37 Pa, a condenser temperature of −55 °C and a heating plate temperature of 20 °C. The drying time was 24 h for the FD reference, and 8 h in the serial combined treatments.The DIC treatment was carried out by using a pilot-scale apparatus (QDPH1021, Tianjin Qin-de New Material Scientific Development Co. Ltd., Tianjin, China) according to previous studies [33,34,35,36]. The DIC treatment was conducted in two stages: in the first stage, the processing vessel was preheated via steel pipes, where steam was injected. Subsequently, the samples were put on wire mesh trays, which were placed on top of the steel pipes in the processing vessel. The samples were heated for 10 min with a processing vessel at 95 °C. In the second vacuum drying stage, the decompression valve was opened to a vacuum chamber to obtain an instant pressure drop (t < 0.2 s) to vacuum (3–5 kPa, absolute pressure), at which the boiling temperature of water is about 24–32 °C. While maintaining this vacuum with a pump, the samples were dried with a processing vessel temperature of 50 °C for 2 h. The processing vessel temperature was maintained via proper steam injection into the steel pipes.Hot air drying (HA): Samples were dried by a hot air drier (DHG-9123A, Shanghai Jinghong Experiment Facility Co., Ltd., Shanghai, China) at 35 °C and 65 °C at an air velocity of 1.2 m/s. The drying times for the single reference treatments were 44 and 16 h for HA35 and HA65, respectively, following the study [37].Heat pump drying (HP) was conducted with a heat pump drier (ZWH-KFY-BT4I/HG, ZhengXu New Energy Technology Co., Ltd., Dongguan, China. The airflow velocity was 2.4 m/s and the relative humidity was below 20% [37].Vacuum drying (VD) was conducted with a vacuum drier (VO200, Memmert, Germany) at 35 °C and 65 °C, respectively. The absolute pressure was 5 kPa for the drying process, which took about 20 min to reach. This pressure was selected so that the oven temperature was above the boiling point of water [38].Infrared radiation drying (IRD) experiments were conducted at 35 °C and 65 °C, respectively, by a laboratory-type short- and medium-wave infrared dryer (STC, Senttech Infrared Technology Co., Ltd., Taizhou, China) [20,39].

### 2.4. Rehydration Method

After drying, the mushrooms were immersed in distilled water at room temperature (25 ± 2 °C) for either 15 min (to determine rehydration rate) or 24 h (to determine volume recovery and WHC). The ratio of the weight of dried mushrooms compared to water was 1:20 (*w*/*w*). At the end of rehydration, these samples were removed from the rehydration water. The excess water was drained off from the mushroom surface before measuring its weight and volume. The remaining rehydration water was analyzed for conductivity and dry matter content.

### 2.5. Quality Indicators of Mushroom

#### 2.5.1. Volumetric Changes of Mushroom after Drying and Rehydration

The volume of fresh, dried, and rehydrated mushrooms was measured by a volume analyzer (VolScan Profiler300, Stable MicroSystem Co., Godalming Surrey GU7 1YL, UK). The Volscan performed 3D imaging, and its software determined the mushroom volume. The volume change of dried and rehydrated mushrooms was expressed relative to the volume of the fresh mushroom. Thus, the volume ratios after drying and rehydration ($\nu_{D}$ and $\nu_{R}$) were calculated via:
$$\nu_{D}=\frac{V_{D}}{V_{0}};\;\nu_{R}=\frac{V_{R}}{V_{0}},$$
where $V_{0}$, $V_{D}$, and $V_{R}$ (mL) were the scanned volumes of fresh, dried, and rehydrated mushrooms, respectively. Measurements were performed in quadruplicate. In the presentation of the results $\nu_{D}$ is denoted as the volume retention ratio, and $\nu_{R}$ is denoted as the volume recovery ratio.

#### 2.5.2. Rehydration Rate

The speed of rehydration is important for the convenience of the consumer. As a measure of the rehydration rate, we determined the change of the weight after 15 min of immersion in the rehydration liquid (tap water at room temperature). This short time was chosen in consideration of the practice of consumers preparing *shiitake* mushrooms and the sensitivity of the measurements. The rehydration rate of dried mushrooms was determined as follows:$$R=\frac{M_{1}-M_{0}}{(M_{0}\times 87\%)*15\;},$$
where $M_{0}$ and $M_{1}$ were sample weights at 0 min and 15 min. 87% was the average dry matter content of dried mushrooms. 15 refers to the 15 min of immersion. The measurements were conducted in quadruplicate.

#### 2.5.3. Dry Matter Loss Ratio of Rehydrated Mushroom

We measured the dry matter loss during rehydration, via collecting the rehydration liquid after the 24 h of rehydration treatment. Both the rehydration liquid and the rehydrated mushrooms were dried in an oven at 105 °C for 24 h (cf. AOAC. 1990) to determine their dry matter contents. The dry matter loss was expressed as the ratio of the weight of dry matter in rehydration liquid to the sum of dry matter in rehydration liquid and rehydrated mushroom. 

#### 2.5.4. Water Holding Capacity 

Water holding capacity is regarded as a measure of the juiciness of mushrooms. In this experiment, we computed the *WHC* of rehydrated samples collected via the data from rehydration experiments. *WHC* was defined as:$$WHC=\frac{W_{r}-W_{d}}{W_{d}},$$
where $W_{r}$ was the weight of the rehydrated mushroom (after removal of surface water), and $W_{d}$ was the dry matter weight of the same rehydrated mushroom without dry matter loss after being dried at 105 °C for 24 h.

#### 2.5.5. Hardness

The hardness of the rehydrated mushrooms was measured by a texture analyzer (Stable Micro System Co., Ltd., Surrey, UK). A cylindrical probe (P/36) was used to compress cubical samples (measuring 0.5 × 0.5 × 0.5 cm^3^). The compression distance on samples was fixed at 50% of their original height. The test speed was 1.5 mm/s. Measurements were performed with 10 repetitions for each sample.

#### 2.5.6. Sensory Evaluation 

Sensory evaluation was conducted by a scaling method [40] where untrained panelists were asked to score the rehydrated samples regarding the intensity of taste and juiciness. We used a 7-point scale with the maximal score representing the highest juiciness or strongest taste. The taste of fresh mushroom was defined as scale 4 and its juiciness was defined as scale 7. 

To simulate practical consumer use, dried mushrooms were rehydrated in boiling water for 5 min and provided to panelists after cooling to room temperature (25 °C). In the sensory tests, we used fresh mushrooms as a reference, which received the same boiling procedure as the dried mushrooms. Because panelists were able to evaluate only a limited number of samples per session, the treated samples were divided into four groups (A–D), as shown in Table 1. In separate sessions, each group, together with the fresh, HA35 and FD references, were evaluated.

### 2.6. Microstructural Properties 

#### 2.6.1. Porosity of Mushrooms after Drying 

The porosity (i.e., the volume fraction of air) of the mushroom was calculated following a simple mass balance, as follows:$$Porosity=\frac{V_{a}-V_{th}}{V_{a}}=1-\frac{\frac{M}{\rho_{th}}}{V_{a}},$$
in this equation $V_{a}$ is the actual volume of mushroom determined by the Volscan, $V_{th}$ is the theoretical volume and $\rho_{th}$ is the theoretical density of the solid/liquid phase of mushroom, which is calculated from its composition estimated via averaging values from various literature sources, as indicated in Table 2. M is the mass of the sample. We assumed the following mass densities of the ingredient [41]: water ($\rho_{w}$ = 1000 kg/L), carbohydrates ($\rho_{c}$ = 1550 kg/L), lipids ($\rho_{l}$ = 920 kg/L), proteins ($\rho_{p}$ = 1330 kg/L), and minerals ($\rho_{m}$ = 2440 kg/L). The theoretical density was then computed from the individual densities following the relation [41]:$$\frac{1}{\rho_{th}}=\underset{i}{\sum }\frac{y_{i}}{\rho_{i}},$$
where $y_{i}$ is the mass fraction of component i.

#### 2.6.2. Porosity after Rehydration 

After rehydration, some air can still remain trapped inside the pores. To quantify this we determined the porosity of rehydrated samples. The method for determining the porosity of rehydrated samples was the same as given in 2.6.1.

#### 2.6.3. Cell Integrity of Rehydrated Mushrooms

Cell membrane integrity was measured indirectly via measuring the electrical conductivity of rehydration liquid by electrolyte conductivity meter (DDS-307A, Leici New Energy Technology Co., Ltd., Shanghai, China). If the cell membrane loses its integrity, one can assume that the conductivity of rehydration liquid has changed significantly compared to tap water. We expressed the conductivity per gram dry matter relative to that of the fresh mushrooms.

### 2.7. Statistical Analysis

All the measurements were performed at least in triplicate (or more if stated otherwise), and the results were presented in the figures by their averaged values and error bars representing standard errors (computed from standard deviations and 95% confidence intervals). Statistical analysis was carried out using SPSS software (Version 18.0, SPSS, Inc., Chicago, IL, USA). Significant differences among product quality attributes were expressed with small letters a–t in the bar graphs showing the results. This was obtained by one-way analysis of variance (ANOVA), for significance at the *p* < 0.05 level, and Duncan’s test. Notably, due to the sensory evaluations conducted in four separated trials, ANOVA comparison of the common references from the four trials was conducted to assure no significant difference between them, and hence the sensory scores of the four trials can be compared together. Furthermore, principal component analysis (PCA) involving all indicators was carried out by employing SIMCA software (Version13.0, Umetrics, Umea, Sweden) to identify clustering within the different treatments, and possible correlations between quality parameters.

## 3. Results

### 3.1. Quality and Microstructure Performance of Dried or Rehydrated Mushrooms with Different Combined Drying Methods 

#### 3.1.1. Volumetric Changes-Comparison of Mushrooms after Drying

The volume retention ratios of mushrooms are shown in Figure 1. The FD and HA65 references show the highest and the lowest volume retention ratio, 0.75 and 0.27, respectively. Among the combined drying methods, DIC-postdrying65 show the lowest volume retention (0.17–0.20), while FD-postdrying35 and 65, show the highest volume retention (0.60–0.69), excluding FD-VD35, but similar to the FD reference (0.75). It is also striking that all pre-drying treatments show poor volume retention. The DIC-postdrying35 shows overall similar volume retention (0.37–0.45) as the HA35 reference (0.41), except for the vacuum drying step: DIC-VD35.

#### 3.1.2. Volumetric Changes-Comparison of Mushrooms after Rehydration

The relative volume recovery after rehydration of all samples is given in Figure 2. Of all dried samples, the highest volume recovery is shown by HA35 (0.95), for which we note that the samples were given 24 h for rehydration. In comparison, the other references HA65 and FD show little volume recovery (0.37,0.52). The combined drying treatments involving FD show lower or similar volume recovery (0.29–0.51) to FD reference (0.52). Surprisingly, the volumes of the rehydrated FD and the FD-post-drying are smaller than shown in Figure 1: the samples have shrunk during rehydration. Amongst the combined dried samples, the best volume recovery is shown by DIC-postdrying35, especially DIC-HA35 (0.74) and DIC-HP35 (0.64). Notice that the volume recovery ratio of the rehydrated fresh mushrooms (which we also treat as a reference) is larger than unity, namely 1.25, meaning that the fresh mushroom swells if immersed in pure water.

#### 3.1.3. Rehydration Rate Comparison of Mushrooms

As shown in Figure 3, FD samples give the fastest rehydration rate (0.34g·g^−1^·min^−1^) amongst all treatments, while HA35 and HA65 show a quite low rehydration rate (0.10, 0.045 g·g^−1^·min^−1^). The result of HA35 is striking, because of its high volume retention after 24 h. However, note that the rehydration rate is based on the volume recovery after the initial 15 min of rehydration. The FD-postdrying35/65 treatments show high rehydration rates, comparable to the FD reference. Further, we note that the rehydration rates of all pre-dried samples are low (0.032–0.11 g·g^−1^·min^−1^), which complies with their low volume recovery, as shown in Figure 2. While DIC-postdrying65 shows quite poor rehydration (0.020–0.028 g·g^−1^·min^−1^), DIC-postdrying35 shows good to moderate rehydration rates. Especially, DIC-HA35 stands out, having a comparable rehydration rate as the FD reference, which is also much higher than the HA35 reference.

#### 3.1.4. Dry Matter Loss Ratio-Comparison of Mushrooms

As shown in Figure 4, after rehydration, the dry matter loss of FD samples is the highest, as high as 50% of its total dry matter content. The dry matter loss of HA35 samples is lowest for all dried samples, accounting for 10% of its total dry matter content, while the dry matter loss is negligible for fresh mushrooms. Among combined drying treatments, DIC-postdrying35/65 shows the lowest dry matter loss, but is comparable to HA65, accounting for 31% of its total dry matter content. For FD-post-drying treatments the dry matter loss is similar to FD reference, but it is slightly lower for pre-drying-FD.

#### 3.1.5. Water Holding Capacity-Comparison of Mushrooms after Rehydration

As shown in Figure 5, HA35 and FD treatments have high WHC, similar to that of fresh mushrooms (0.95), but also comparable to several combined drying treatments, namely FD-post-drying and DIC-postdrying35. Poor WHC is shown by DIC-postdrying65 and predrying65-DIC treatments, varying from 0.70–0.79.

#### 3.1.6. Hardness-Comparison of Fresh and Rehydrated Mushrooms

As shown in Figure 6, many of the treatments show similar hardness (after rehydration) as the fresh mushrooms (433 g), namely FD-post-drying, predrying35-FD and DIC-postdrying35 (except DIC-VD35). The highest values of hardness are shown by DIC-postdrying65, and several treatments within pre-drying-DIC, varying from 1295 g to 2018 g.

#### 3.1.7. Sensory Evaluation of Mushrooms after Rehydration

The sensory scores of the panelists are listed in Appendix A. As illustrated above, the fresh mushroom was taken as the anchor for all the other evaluated samples. Among the shared references, HA35 is shows similar taste as the fresh mushroom, but the FD samples show a lower rating for taste. HA35 and FD references show similar juiciness, which is lower than that of the fresh mushroom. Due to the large variations in the sensory scores amongst panelists, it is difficult to draw conclusions directly from the tables in Appendix A. We rely on the PCA analysis to obtain trends from the sensory evaluation.

#### 3.1.8. Porosity after Drying

As shown in Figure 7, the porosity of all dried samples increases when compared to that of fresh ones (0.61). FD gives the highest porosity (0.94), while HA65 gives the lowest (0.76). For both DIC and FD treatments, it holds that combining with post-drying gives higher porosity than combining with pre-drying. We observe that, for DIC treatments, this porosity enhancement only remains when the post-drying temperature is lower (35 °C) and the porosity of DIC-postdrying65 is the lowest amongst the combined drying treatments, and similar to that of the fresh mushroom. FD-post-drying treatments show a porosity as high as the FD reference.

#### 3.1.9. Porosity after Rehydration

The porosity after rehydration entails pores where air is entrapped. As explained below, we will regard the porosity of rehydrated samples as a measure of the interconnectivity of the dried mushroom. For samples with high porosity, and thus much entrapped air, we assume a low interconnectivity. According to Figure 8, the porosity of rehydrated FD reference samples is the lowest (0.11), corresponding with the highest interconnectivity. Among the combined drying treatments, samples dried by FD-postdrying35/65 show the lowest values of porosity (0.19–0.25), but only slightly lower than DIC-postdrying35 (0.21–0.29). The porosity of the pre-drying involved with DIC samples is overall higher than that for post-drying samples, with the worst performance by DIC-postdrying65 (0.32–0.37).

#### 3.1.10. Cell Membrane Integrity

It is shown in Figure 9 that the relative conductivity of rehydration liquid of all dried samples is higher than 1, indicating that all drying methods cause damage to the cell membrane. The lowest and highest relative conductivity ratios are shown by the HA35 and FD references, respectively. Among the combined drying methods, DIC-postdrying35 performs best with relative conductivity ratio only slightly higher than HA35. Most FD combined treatments perform better ratios than FD reference, excluding FD-postdrying65. The DIC-treatments, other than DIC-postdrying35, show similar behavior to these FD combined treatments.

### 3.2. PCA Analysis

The loading plot and the score plot of the PCA analysis are shown in Figure 10 and Figure 11. PC1 and PC2 explain together 73% of the total variance. The loading plot has shown that many of the quality indicators have a high degree of correlation. PC1 is positively correlated with WHC, juiciness, rehydration rate, and volume recovery after rehydration, and is negatively correlated with hardness. These quality indicators are indeed related, namely to juiciness and texture. PC2 is positively correlated with dry matter loss, relative conductivity of the rehydration liquid, and slightly (negatively) correlated with taste. The porosity indicators depend on both PC1 and PC2.

As shown by the score plot in Figure 11, we observe a large difference between the single-drying methods, i.e., the references FD and HA35, which can be considered as two extremes in drying performance, as we have discussed in the introduction. Furthermore, we observe that there is a large degree of clustering amongst the combined drying methods. Thus, the clustering is probably due to the fact that the type of conventional drying method (HA, HP, IR, VD) had limited impact on the quality indicators in general, except for the DIC-postdrying35 treatments. 

In Figure 11 we have indicated the following clusters:(a)FD-postdrying35/65(b)predying35/65-DIC+predrying35/65-FD(c)DIC-postdrying65(d)DIC-postdrying35

From the results of clustering, it can be seen that cluster (a) FD-postdrying35/65 is very close to the FD reference, giving a high volume retention after drying, a high rehydration rate, but accompanied by a high dry matter loss. Cluster (b) is located to HA65, giving low quality performance such as a low volume retention after both drying and after rehydration, a quite low rehydration rate, but a moderate dry matter loss. The cluster (c) DIC-postdrying65 also gives poor performance similar to cluster (b), albeit with lower rehydration rate, and lower loss of dry matter. Compared with the other combined drying methods, the cluster (d) DIC-postdrying35 shows improved quality, combining good volume recovery and a fast rehydration rate, but with a low dry matter loss. Its response is about midway between references HA35 and FD, as has been our target for the combined drying method.

## 4. Discussion

Due to the large amount of generated data, we use the results of the PCA analysis to guide our discussion of the combined drying methods. The PCA analysis showed a strong clustering amongst the combined drying methods, and consequently we have performed our discussion mainly on the level of these clusters. For each cluster, we explained the observed differences in quality indicators in terms of microstructural changes. Before that, we list some generally accepted hypotheses relating quality to microstructure, which are used in the following discussion. 

Among all drying methods, extreme differences were found in different indicators. HA35 gives the highest volume recovery ratio and the least dry matter loss. FD performs the largest volume retention ratio and rehydrates fastest, but it shows the most dry matter loss value. DIC-postdrying65 treatments leads to the least volume retention ratio, volume recovery ratio, rehydration rates and WHC value, but gives the highest hardness. The underlying mechanism for the difference can be explained by microstructure characteristics.

High rehydration rates are explained by high permeability, which is attributed to both a high porosity (after drying) as well as a high interconnectivity [46]. The porosity after rehydration can be viewed as a measure of this interconnectivity, air bubbles trapped during rehydration determining the (final) porosity of rehydrated samples. The trapped air bubbles have not escaped due to the poor interconnectivity [10,47,48]. In our study, the high interconnectivity between pores in dried samples like FD combined treatments is corroborated by (1) the high rehydration rate, whose magnitudes are similar to those of the single FD treatment, and (2) the low porosity after rehydration. 

Poor rehydration rates are generally attributed to (pore) collapse of tissue and subsequent case hardening [49]. Recent literature on soft porous polymer gels [50] relates the prevention of pore collapse during drying to interconnectivity between pores, which is shown earlier. The collapse of water-filled pores occurs only when they are not interconnected. Similarly, we assume water-filled cells in vegetables will collapse if a porous structure with sufficient interconnecting channels is present. The size of these channels defines the capillary pressure, and if the capillary pressure is smaller than the elastic modulus of the tissue, collapse does not occur [50].

The loss of cell membrane integrity results in dry matter loss, i.e., the leaching of solutes from the vacuole. During freeze-drying one can expect that all cell membrane integrity is lost due to ice crystals puncturing the cell membrane [5], or due to the high osmotic pressures—leading to osmotic lysis [51]. This leads to high leaching of solutes into the rehydration water and high (relative) conductivity of the rehydration water. Besides, solute leaching is found to be independent of the microstructure (remaining after the loss of cell membrane integrity) of the dried vegetable [10]. In contrast, fresh mushrooms without cell damage give negligible dry matter loss and notably grow 1.25 times larger in volume, meaning cell integrity is also critical for high volume recovery ratio [14]. 

For cluster (a) FD-postdrying35/65, the FD treatment before post-drying can help to improve the volume retention after drying (except FD-VD35) and the rehydration rate when compared with HA35/65 dried samples. The results of improved volume retention and rehydration rate can be attributed to the higher porosity and interconnectivity created by the freeze-drying pretreatment (shown in Figure 7 and Figure 9). The porosity values are similar to that of the single FD treatment. Notably, although the duration of the FD pretreatment (8 h) is shorter than the single FD drying (24 h), it is sufficient to create similar porosity. Apparently, ice sublimation finishes within the first 8 h of the (single) FD treatment. The porous structure is thus already created at the end of the ice sublimation, enhancing the mass transfer in the subsequent drying in the vacuum stage of FD. The created interconnected, porous tissue prevents the subsequent collapse of the structure during the post-drying, as commonly happens during hot air drying, like in the HA35 or HA65 reference treatments. Similar prevention of collapse by a freeze-drying pretreatment is also demonstrated for serial combined drying applied to mushrooms and carrots [52,53]. 

Furthermore, we note that the dry matter leached during rehydration of FD treated samples (50% on dry weight basis) is larger than the amount of solutes from the vacuole (which is about 30% on a dry weight basis; see Table 2). The loss of cell membrane integrity is thought to happen already in the freezing step, before the actual (freeze) drying via ice sublimation. Hence, one can indeed expect similar dry matter loss for FD-post-drying as for the single FD treatment. As illustrated above, we think that dry matter loss is also partly due to the damaged or fractured cell wall material. The extent of the damage to the cell wall material has probably led to the low hardness. 

FD-postdrying35/65 gives a lower volume recovery than HA35, but similar to that of single FD treatments. If we compare the sample volume after drying with those after rehydration, we observe a decrease of volume due to rehydration similar to single FD treatments. This reduction of volume during rehydration is hardly observed in the literature and, therefore, it is an important finding. This reduction of volume could be due to the mechanical stresses, frozen-in during freeze-drying, which are dissipated via relaxation of the stretched cell wall material, leading to further shrinkage. 

We attribute the high WHC and juiciness to the large amount of water contained in the pores after rehydration, which is pressed out easily during eating. The latter can be attributed to the high permeability of FD pretreated samples, which also facilitates the high rehydration rate.

For cluster (b) predrying35/65-FD/DIC, poor product quality is observed, comparable to the HA65 reference. Similar to that found by Siebert [53], the microstructural damage imparted by pre-drying cannot be undone by pore-enhancing treatments, i.e., FD or DIC. The pre-drying samples showed low volume retention, and if compared to the volume after drying, one observes there is little water uptake by the pre-drying samples. This is also indicated by the low rehydration rate. Furthermore, there is little connectivity between pores as indicated by the high porosity after rehydration. As stated above, collapse happens if there is little interconnectivity between pores. In line with the poor rehydration, the pre-drying samples show low WHC, low juiciness, and high hardness. 

Unexpectedly, the pre-drying-DIC samples performed as badly as the pre-drying-FD samples. Commonly, pre-drying is a required pretreatment before puffing via the DIC treatment [54]. In our case, the volume expansion did not happen, probably due to the severity of case hardening caused by our pre-drying treatment. The compacted structure had probably a too high hardness, which could not be overcome by the pressure drop induced by DIC [55]. 

For cluster (c), the DIC-postdrying65 treatments show poor performance in hydration properties, such as low volume recovery, low rehydration rate, low WHC, and high hardness. Regarding this, it scores even worse than the HA65 references. On the other hand, their dry matter loss is lower than the FD treatments, but comparable to HA65 and DIC-postdrying35. Remarkably, their volume retention after drying is the lowest of all treatments. Their poor performance is in stark contrast to DIC-postdrying35, which does show good hydration properties. Hence, the poor performance of DIC-postdrying65 must be sought in the combined effect of the high drying temperature and the long duration of the postdrying65 treatment. If we include the 2 h of the vacuum drying stage of the DIC treatment, the total drying time is similar or even longer than that of the HA65 reference. The severity of the postdrying65 treatment is indicated by the lowest volume retention, and lowest porosity after drying (Figure 7). The high drying temperature leads to a higher loss of cell membrane integrity, as indicated by the higher relative conductivity ratio as compared to DIC-postdrying35. Besides, the postdrying65 might lead to strong case hardening, resulting in very little and slow water uptake during rehydration. Any benefit of the DIC treatment, which is present as shown by the performance of DIC-postdrying35, has been eliminated by the severity of the postdrying65 treatment. 

Unexpectedly, cluster (d), the DIC-postdrying35 treatments, shows a good performance, namely a high volume recovery ratio and a high rehydration rate, but a low loss of dry matter and cell membrane integrity. The volume recovery is similar to the HA35 reference, but higher than the FD reference, while the rehydration rate is much higher than the HA35, and in the case of DIC-HA35 even higher than the FD reference. The dry matter loss is lower than FD, comparable to that of HA65, but lower than HA35. Within this cluster the DIC-HA35 has shown the best performance. The DIC-HA35 treatment combines the positive traits of freeze-drying (FD) treatments with the positive traits of mild hot air drying (HA35), which is indeed the objective of this study. Furthermore, as shown in Table 1, the total processing time of the serial combined drying of DIC-HA35 (3 + 33 h) is significantly shorter than the HA35 reference (44 h). 

The stark difference in performance between DIC-postdrying35 and DIC-postdrying65 is likely to be due to the prevention of severe case hardening, i.e., collapse of the surface layer. This assumption is supported by the moderate volume shrinkage and high porosity after drying, the good volume recovery after rehydration (even better than FD), good rehydration rates (comparable to FD), and reasonably low porosity after rehydration for DIC-postdrying35. Earlier, we have explained the prevention of collapse in terms of good interconnectivity between pores. The high rehydration rate and low porosity after rehydration indicate that this indeed the case. This prevention of collapse is quite surprising, compared to the performance of HA35 and DIC-postdrying65. HA35 does have a good volume recovery, but its rehydration rate is quite poor, indicating a low permeability and interconnectivity. We think that low permeability and interconnectivity of HA35 especially holds in the surface layer, as the (average) volume shrinkage and the porosity after drying and after rehydration of HA35 is similar to DIC-HA35. The surface layer of DIC-HA35 has somehow retained the good permeability and interconnectivity one can find in fresh mushrooms. 

It is not likely that the DIC-pretreatment acts on the porosity of the mushroom. Our measurements have shown that the moisture content decreases only minutely from 82% to 80% during the DIC-pretreatment. For the creation of a high porosity the food material needs to be pre-dried, as shown by Allaf [56]. We think in our case that the DIC-pretreatment acts at the level of the interconnectivity of pores. During the puffing stage of the DIC treatment, there is an instant drop in pressure of about 1 bar. The gas inside the mushroom can probably not escape directly and it will expand due to the lower ambient pressure. Consequently, the gas expansion increases the volume of the mushroom and the size of interconnecting channels between pores. Subsequently, the (expanded) mushroom will be subjected to vacuum drying for 2 h. The mass transport during this vacuum drying is driven by pressure differences, similar to the vacuum drying of wood [57]. The pressure in the mushroom remains at the saturated vapor pressure at the governing product temperature (which is about 40 degrees due to evaporative cooling), thus retaining the pressure gradient for convective mass transport. Due to the high initial porosity of the fresh mushroom, the dehydration during the 2 h vacuum drying of DIC treatment happens throughout the whole tissue of the mushroom, in contrast to HA35—where dehydration happens mainly at the surface layer in the first 2 h of drying. Besides, the increase in atmospheric pressure at the end of the DIC treatment is performed slowly such that the volume of the mushroom does not implode. Hence, the advantageous microstructural properties are retained throughout the DIC treatment. 

The same DIC treatment has been performed in both the DIC-postdrying35, as well as the DIC-postdrying65. While post-drying at 65 °C destroys the advantageous microstructure, post-drying at 35 °C can still retain it. Comparison of HA65 with HA35 indicates that drying at 65 °C is more severe leading to lower volume shrinkage, lower volume recovery, lower rehydration rates, lower porosity after drying, and a high porosity after rehydration (indicating the lower interconnectivity). Furthermore, the relative conductivity and dry matter loss in HA35 is much lower than HA65. Thus, the cell membrane integrity is better preserved during drying at 35 °C, which also happens during the DIC-post-drying as shown by the difference in relative conductivity between DIC-postdrying35 and DIC-postdrying65. If cell membrane integrity is (partially) retained, the turgor pressure is also (partially) retained. The turgor pressure means that the effective elastic modulus is higher than that of the mushroom dried at 65 °C, where turgor pressure is lost. The higher elastic modulus retained by the low drying temperature combined with the better interconnectivity caused by the DIC treatment is the probable reason for the prevention of collapse during the post-drying at 35 °C, which does happen at the post-drying at 65 °C. Additionally, if the cell membrane loses integrity, intracellular water is pressed out to the extracellular space (pores and interconnecting channels) by the relaxing (previously stretched) cell walls. Due to capillarity effects, this water will predominantly be captured by the small interconnecting channels, rather than the large pores. Loss of membrane integrity will first happen at the surface layer in the DIC-HA65, thus closing the interconnection of (air-filled) pores with the environment. This loss of interconnectivity leads to the collapse of the surface layer. During DIC-HA35, loss of cell membrane integrity happens much less, and probably at the late stage of the post-drying treatment. Hence, the collapse of the surface layer is probably prevented. Future research will have to substantiate our hypothesis.

## 5. Conclusions

The changes of quality and microstructure properties of mushrooms treated by serially combined drying with FD and DIC as pore-creation methods were analyzed. Among the serially combined drying treatments, there are only a few treatments which perform better than HA and FD references. The observed behavior of the different treatments has been explained in terms of changes of microstructure during drying and rehydration. 

The best performance is shown by DIC-HA35 dried samples which show high volume recovery like that of HA35, but with a much shorter drying time; they can rehydrate as fast as the FD dried samples but with less dry matter loss. The unexpectedly good performance could be due to the retention of pore interconnectivity during the DIC-HA35 treatment, resulting from (a) the slight expansion of structure during DIC, and (b) the retention of cell membrane integrity, avoiding extracellular water filling the interconnecting channels between pores. 

On the other hand, FD or DIC after drying cannot undo the damage imparted by pre-drying. No advantage is found for FD-post-drying over FD, which is probably due to the long duration of the freezing step. Quality might be improved via shorter freezing time. Nevertheless, we are inclined to select DIC-HA35 as the combined drying method in our further research, as it has been shown to improve the quality performance of rehydrated mushrooms when compared to single convective drying methods. 

In future study, we think it is worthwhile investigating the DIC-HA35 serial combined drying method for (a) the validation of our hypothesis, and (b) the optimization of drying of mushrooms. Besides, whether the DIC-post-drying is beneficial to other vegetables or fruits remains to be seen.

## Figures and Tables

**Figure 1 foods-10-00769-f001:**
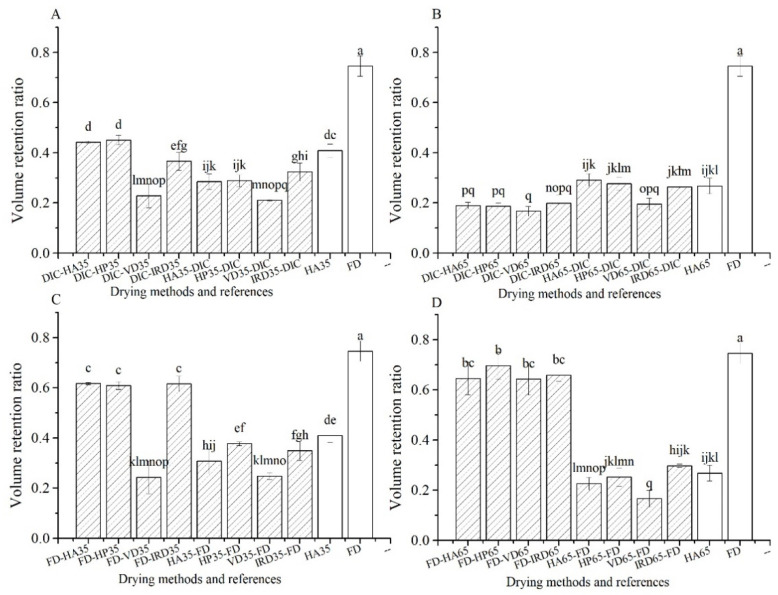
Volume retention ratio of mushrooms, directly after drying, for the different combined drying methods (
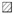
) and reference methods (□). (**A**–**D**): group names as shown in Table 1.

**Figure 2 foods-10-00769-f002:**
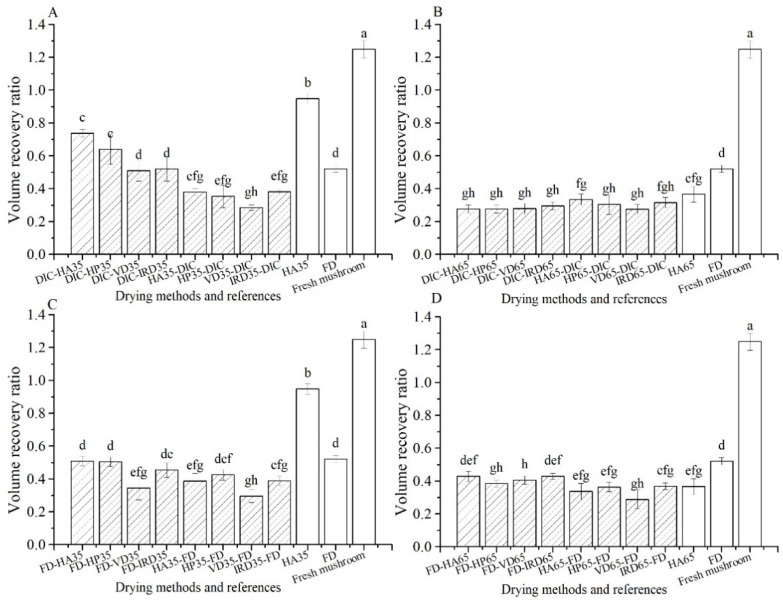
Volume recovery ratio of mushrooms after rehydration with different drying methods (
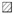
) and references (□). (**A**–**D**): group names as shown in Table 1.

**Figure 3 foods-10-00769-f003:**
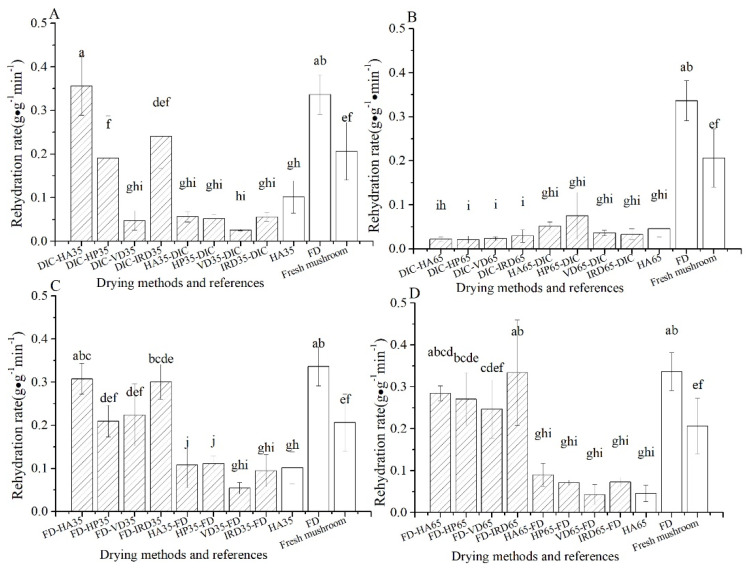
Rehydration rate of mushrooms with different drying methods (
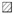
) and references (□). (**A**–**D**): group names as shown in Table 1.

**Figure 4 foods-10-00769-f004:**
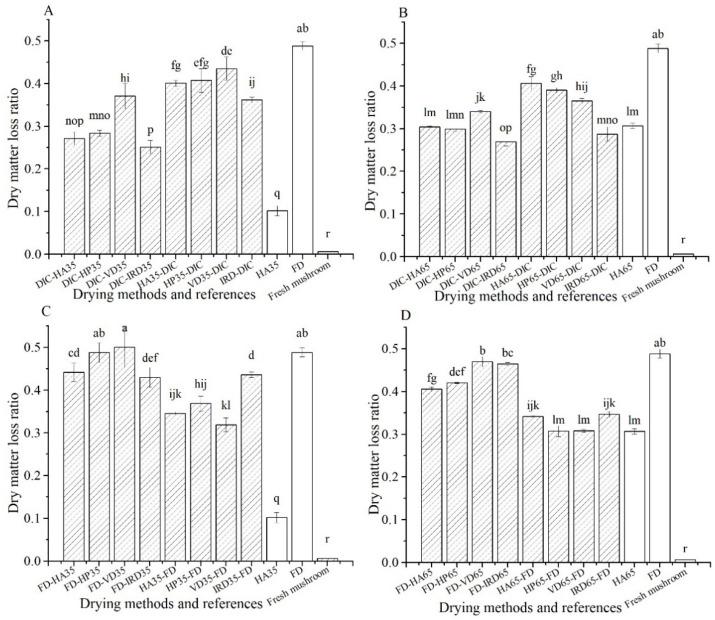
Dry matter loss ratio of rehydrated mushroom with different drying methods (
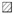
) and references (□). (**A**–**D**): group names as shown in Table 1.

**Figure 5 foods-10-00769-f005:**
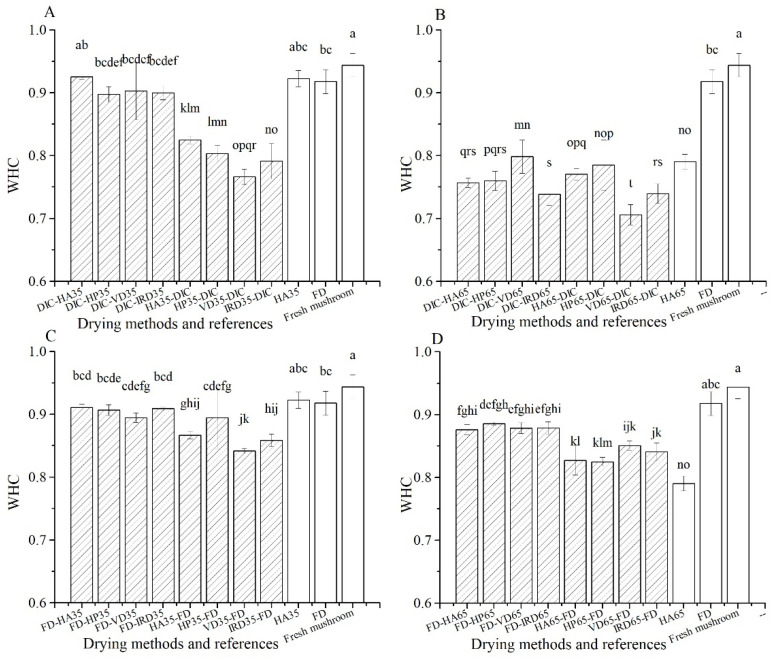
Water holding capacity (WHC) of rehydrated mushrooms with different drying methods (
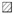
) and references (□). (**A**–**D**): group names as shown in Table 1.

**Figure 6 foods-10-00769-f006:**
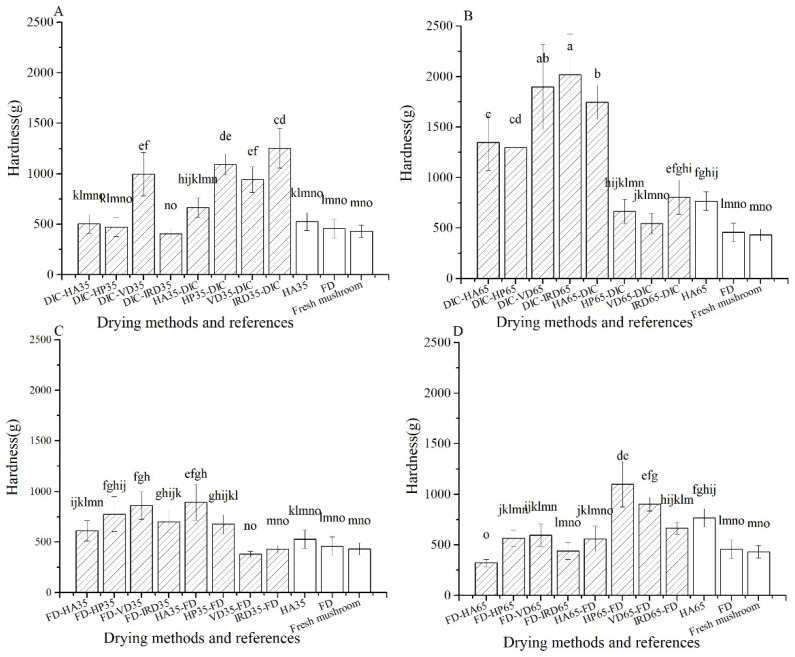
Harness of rehydrated mushrooms with different drying methods (
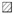
) and references (□). (**A**–**D**): group names as shown in Table 1.

**Figure 7 foods-10-00769-f007:**
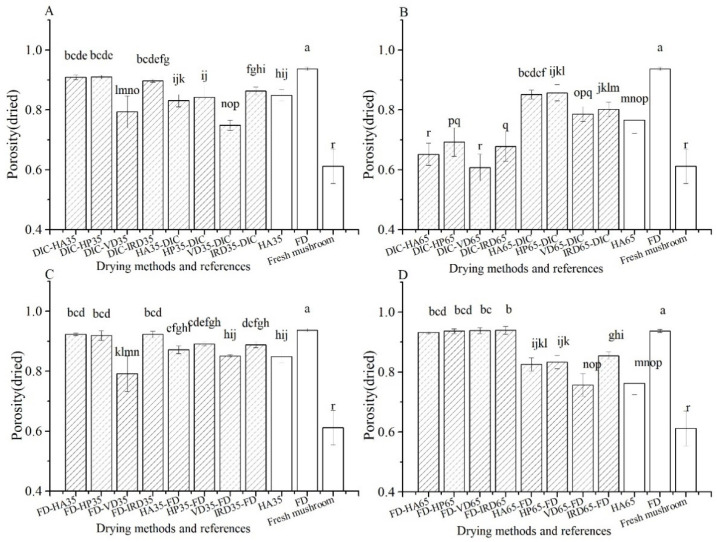
Porosity of dried mushrooms treated with different drying methods (
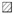
) and references (□). (**A**–**D**): group names as shown in Table 1.

**Figure 8 foods-10-00769-f008:**
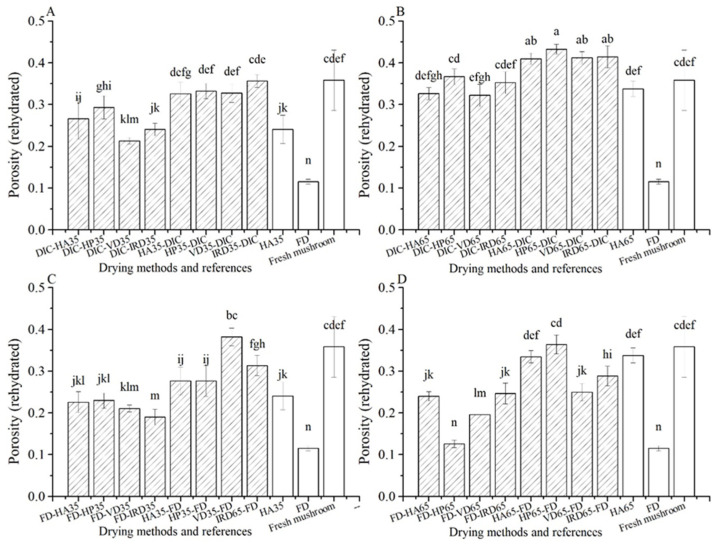
Porosity of rehydrated and fresh mushrooms treated with different drying methods (
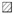
) and references (□). (**A**–**D**): group names as shown in Table 1.

**Figure 9 foods-10-00769-f009:**
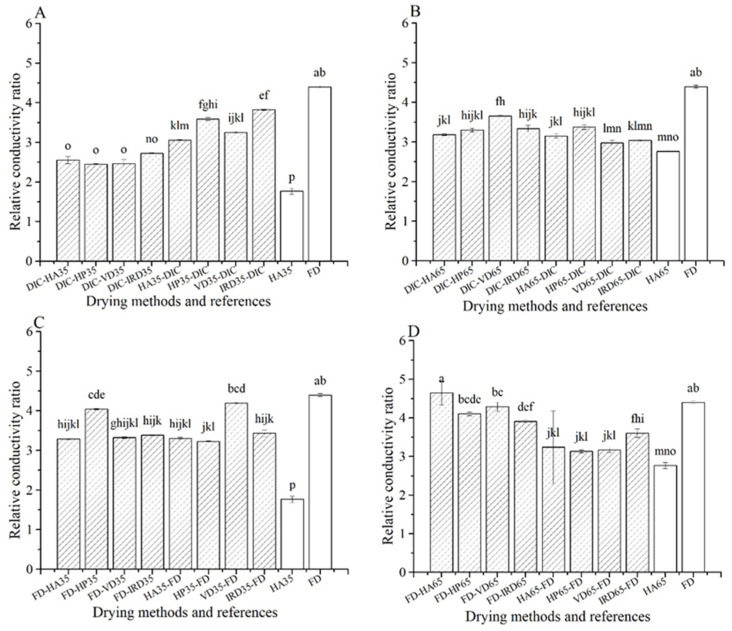
Relative conductivity ratio of rehydration liquid with different drying methods (
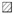
) and references (□). (**A**–**D**): group names as shown in Table 1.

**Figure 10 foods-10-00769-f010:**
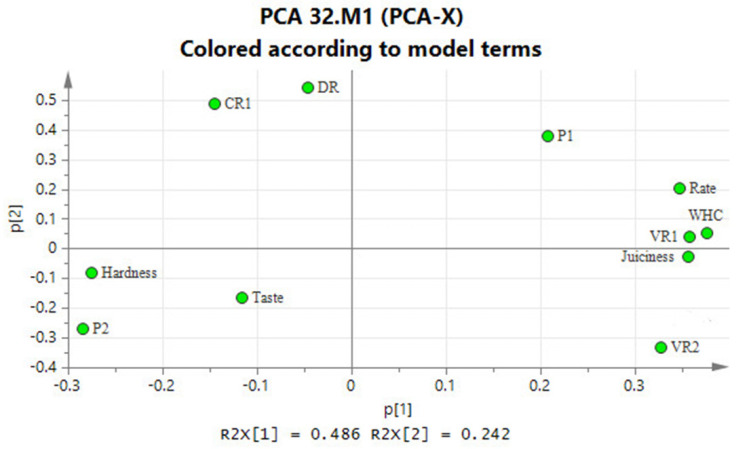
Correlation loadings plot from PCA on mean-centered data. VR1: volume ratio of dried samples, VR2: volume ratio of rehydrated samples, P1: porosity of dried samples, P2: porosity of rehydrated samples, Rate: rehydrate rate; CR1: relative conductivity ratio DR: dry matter loss ratio.

**Figure 11 foods-10-00769-f011:**
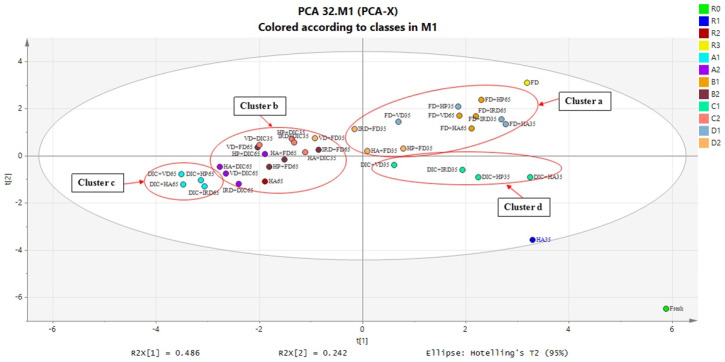
Score plot of the products’ treatments and references (listed in Table 1) showing clustering of the following treatments: cluster a, FD-postdrying35/65; cluster b, predying35/65-DIC+predrying35/65-FD; cluster c, DIC-postdrying65; cluster d, DIC-postdrying35.

**Table 1 foods-10-00769-t001:** Overview of the combined drying methods with applied temperature and drying time.

	Combined Methods	Instant controlled Pressure Drop Drying (DIC) Combined			Freeze Drying (FD) Combined	
Group Name /Temperature		A	B		C	D
		35 °C	65 °C		35 °C	65 °C
DIC-post-drying		DIC-HA35 (33 h)	DIC-HA65 (13 h)	FD-post-drying	FD-HA35 (20 h)	FD-HA65 (15 h)
		DIC-HP35 (31 h)	DIC-HP65 (13 h)		FD-HP35 (18 h)	FD-HP65 (14 h)
		DIC-VD35 (43 h)	DIC-VD65(27 h)		FD-VD35 (26 h)	FD-VD65 (18 h)
		DIC-IRD35 (27 h)	DIC-IRD65 (10 h)		FD-IRD35 (17 h)	FD-IRD65 (13 h)
DIC-pre-drying		HA35-DIC (41 h)	HA65-DIC (16 h)	FD-pre-drying	HA35-FD (46 h)	HA65-FD (21 h)
		HP35-DIC (38 h)	HP65-DIC (13 h)		HP35-FD (43 h)	HP65-FD (18 h)
		VD35-DIC (49 h)	VD65-DIC (25 h)		VD35-FD (53 h)	VD65-FD (30 h)
		IRD35-DIC(33 h)	IRD65-DIC (12 h)		IRD35-FD (38 h)	IRD65-FD (17 h)

HA35,HP35,VD35,IRD35 means hot air drying, heat pump drying, vacumming drying, infrared drying at 35 °C, HA65,HP65,VD65,IRD65 means hot air drying, heat pump drying, vacumming drying, infrared drying at 65 °C.

**Table 2 foods-10-00769-t002:** Fraction of ingredients in dry matter of shiitake mushrooms.

Number	Carbohydrates	Soluble Carbohydrate (Mannitol+ Trehalose)	Insoluble Carbohydrate (Fiber)	Protein *	Fat	Ash(K_3_PO_4_)	References
1	0.82	0.36	---	0.13	0.01	0.042	[42]
2	---	---	0.28	0.17	---	0.063	[43]
3	0.71	---	0.46	0.14	0.031	0.072	[44]
4	0.87	0.13	---	0.044	0.017	0.067	[45]
Average	0.72 ± 0.14			0.17 ± 0.11	0.024 ± 0.007	0.067 ± 0.007	

* In all these data sources, protein content is determined with a nitrogen-to-protein conversion factor of 4.38, which is required for correction of nitrogen-containing chitin.

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
