# Peer review of "Study on the Rehydration Quality Improvement of shiitake Mushroom by Combined Drying Methods"

_foods, 2021, doi:10.3390/foods10040769_

Round 1
Reviewer 1 Report
The manuscript needs to be improved according to comments:
Comment 1
The purpose of the work should be made clearer in the introduction!
Comment 2
Did the authors carry out a method for planning the experiment? Was drying and rehydration parameters optimized?
Comment 3
Why did the authors not study the color?
Comment 4
What was the initial humidity of the tested materials?
Was the material uniform for all drying methods?
Comment 5
Figures 1-7 are illegible. They require modification! Identify the methods (3-4) that show the extreme differences and describe them in the discussion!
Comment 6
The description in Figure 11 is illegible?!
Comment 7
Literature related to the drying and rehydration process should be added:
- Multi-objective optimization of the apple drying and rehydration processes parameters. Emir. J. Food Agric. EJFA 2018, 30, 1–9.
- Apple Cubes Drying and Rehydration. Multiobjective Optimization of the Processes. Sustainability 2018, 10, 4126.
Reviewer 2 Report
Shiitake mushroom is considered as a popular food by the consumers. Development of processes for efficient drying producing high quality product can provide useful information not just for the science but also for practice. Investigation of product structure exposed to different dehydration processes is important to assess the effects of drying methods on product quality.
Therefore, the topic of manuscript foods-1157753 has high relevance and can be considered as interesting for the readers.
Manuscript is general well structured. Introduction section is a good summary of the state-of-art. Research motivations are well defined.
Materials and methods are adequate to the main objectives of the research. Methods (drying methods and methods for the investigation of quality parameters) are described clearly.
Manuscript contains significant and interesting finding. Results are valuable for the science and practice, as well.
Comments, suggestions:
It is not clearly defined why used the authors the temperature of 35 and 65 Celsius for the drying experiments. Please add this information to the manuscript.
In sections 3.1.1.-3.1.9 please add data for the discussion part (experimental/calculation data for ’lowest volume retention’, ’similar volume recovery’, ’ rehydration rates of all predried samples are low’, ‘poor WHC is shown by DIC-postdrying65‘etc).
I suggest the authors to discuss briefly the industry scale applicability of the different drying methods and the energy efficiency, as well.
